# Warburg-Cinotti disease variant p.Tyr740Cys enhances catalytic activity of DDR2 kinase

Ziteng Hao[ID]¤, Birgit Leitinger[ID]*

National Heart and Lung institute, Imperial Collage London, London, United Kingdom

¤ Current address: Randall Centre for Cell & Molecular Biophysics, King's College London, London, United Kingdom

* b.leitinger@imperial.ac.uk

## Abstract

The discoidin domain receptor DDR2 is a collagen-binding receptor tyrosine kinase whose dysregulation is associated with a wide range of diseases. Missense mutations in the DDR2 kinase domain cause Warburg-Cinotti syndrome in an autosomal dominant manner. Warburg-Cinotti syndrome is a severe connective tissue disorder, characterised by a range of manifestations including joint contractures of the hand, corneal vascularisation and pannus, skin fusion and infection, keloid plaques and acro-osteolysis. The Warburg-Cinotti variants, p.Leu610Pro and p.Tyr740Cys, were previously hypothesised to cause disease through a gain-of-function mechanism but mechanistic studies addressing this notion have been lacking. Here we show that both disease variants exhibit ligand-independent constitutive autophosphorylation when expressed as full-length proteins in mammalian cells. We also characterised the enzyme kinetics of soluble WT and DDR2-Y740C kinase constructs. WT DDR2 kinase was found to follow the same two-step activation mechanism previously characterised for DDR1 kinase but with enhanced autophosphorylation and substrate phosphorylation rates. Compared with WT DDR2, DDR2-Y740C displayed further enhanced autophosphorylation and substrate phosphorylation rates, but no effect on ATP binding affinity. The increased catalytic rates of unphosphorylated DDR2-Y740C kinase were similar to those of fully phosphorylated WT DDR2, indicating that the missense variant bypasses all autoinhibitory constraints and adopts the fully active kinase conformation. Tyrosine-740 is a residue in the A-loop of DDR2 kinase that forms autoinhibitory hydrogen bonds with key catalytic residues. These hydrogen bonds cannot form in the cysteine-substituted variant, providing a structural explanation for the release of the A-loop from its autoinhibitory conformation.

**Data availability statement:** All relevant data are within the manuscript and its Supporting Information files.

**Funding:** BBSRC grant BB/R006245/1 (BL), www.ukri.org/councils/bbsrc. This study was also partly supported by an Eric Ivor Hopton Endowment to the National Heart and Lung institute, Imperial College London. The funders had no role in study design, data collection and analysis, decision to publish or preparation of the manuscript.

**Competing interests:** The authors have declared that no competing interests exist.

## Introduction

Cells interact with their environment through cell surface receptors, which control many aspects of an organism's physiology. Dysregulated receptor function can result in a wide variety of human diseases, affecting diverse organs and body systems. Warburg-Cinotti syndrome is a rare connective tissue disease characterised by pathological features that include blepharophimosis, progressive corneal vascularisation, deafness, acro-osteolysis, excessive contractures, and keloid-like plaques [1–3]. The syndrome was found to be caused by either of two heterozygous missense variants of the receptor tyrosine kinase (RTK) discoidin domain receptor 2 (DDR2) [3].

RTKs are single-span transmembrane (TM) proteins that respond to ligand binding by phosphorylating their intracellular kinase domain and sometimes also its flanking regions, which in turn stimulates downstream pathways [4]. The 19 subfamilies of human RTKs show diversity in their ligand-binding extracellular domain (ECD) architecture while containing a highly conserved intracellular catalytic tyrosine kinase domain (TKD) [4–6]. The RTK TM domains are connected to the ECDs and TKDs by more flexible extracellular and intracellular juxtamembrane (JM) regions.

To prevent aberrant signalling, the RTK catalytic activity normally is tightly controlled in the absence of ligands by several auto-inhibitory mechanisms [7,8]. Ligand binding leads to RTK dimerisation, conformational change, or oligomerisation which brings the TKDs into close proximity, thereby facilitating *trans*-phosphorylation of tyrosine residues in the TKD and flanking cytoplasmic regions, which in turn relieves the autoinhibition [9,10]. TKDs consist of a smaller N-lobe and a larger C-lobe, enclosing the active site in the cleft between the lobes [11]. Active kinases share a common catalytically competent conformation, in which the regulatory 'αC helix' in the N-lobe moves to a 'C-IN' conformation, thereby enabling optimal ATP positioning. Another important catalytic regulatory element is the activation loop (A-loop), which contains a conserved Asp-Phe-Gly (DFG) motif at its N-terminus and several tyrosine residues serving as phosphorylation sites [12]. The DFG motif is involved in coordinating ATP, while tyrosine phosphorylation allows the A-loop to adopt an extended conformation at its C-terminal portion, which provides a site for binding the peptide substrate [13,14]. In most kinases, the unphosphorylated A-loop suppresses kinase catalytic activity, but specific inhibition mechanisms can vary [15,16].

The DDRs form a small subfamily of RTKs, consisting of DDR1 and DDR2, that function as receptors for different types of collagens, thereby linking RTK signalling to the extracellular matrix (ECM) [17–19]. The DDRs undergo ligand-induced activation (autophosphorylation) with unusually slow kinetics [4,17,18]. In the absence of ligand, they exist as constitutive noncovalent dimers on the cell surface [20–22]. Ligand binding leads to higher-order DDR clustering (oligomerisation), which results in *trans* autophosphorylation between neighbouring dimers [23–25].

The DDRs contain two globular domains in their extracellular region: an N-terminal discoidin (DS) domain, connected to a DS-like domain [26,27]. A poorly conserved extracellular JM segment links the DS-like domain with the TM domain [27]. The intracellular region starts with a long, flexible JM segment, followed by the C-terminal TKD [19,28]. The

last quarter of the intracellular DDR1 JM region (residues 566–591), termed the 'JM4' segment, was found deeply inserted into the kinase active site in inactive DDR1, thereby providing an additional inhibitory control element to the A-loop [29]. The JM4 segment is highly conserved between DDR1 and DDR2 (with 23/26 amino acids being identical), indicating a similar kinase activation mechanism for both DDRs. However, the rest of the intracellular JM region (123 amino acids in the DDR1b isoform, 94 amino acids in DDR2) is less well conserved; hence, it is important to experimentally verify the DDR2 activation mechanism.

Aberrant DDR signalling contributes to the disease progression of multiple human diseases, such as osteoarthritis, fibrotic disease, and many types of cancer [30–33]. In 2018, Xu et al. identified two heterozygous missense variants (c.1829T > C; p.Leu610Pro and c.2219A > G; p.Tyr740Cys) located within the DDR2 TKD as being responsible for Warburg-Cinotti syndrome [3]. The authors suggested these variants to be gain-of-function mutations associated with enhanced DDR2 kinase activity and signalling [3]. However, experimental evidence for DDR2-L610P and DDR2-Y740C exhibiting enhanced activity has been lacking.

In this study, we addressed the hypothesis that the DDR2 mutations relieve autoinhibition and enhance kinase activity. To this end, we expressed and purified soluble recombinant DDR2 kinase constructs (DDR2-WT and DDR2-Y740C) that included the regulatory JM4 segment and determined their autophosphorylation characteristics and kinase catalytic parameters. The purified kinase construct of the DDR2-L610P variant could not be obtained due to issues with protein solubility. Biochemical experiments demonstrated that the DDR2 kinase undergoes the same stepwise activation process as DDR1, albeit with much quicker autophosphorylation rates than the DDR1 kinase. The data further show that A-loop tyrosine substitution (Y740C) releases autoinhibition of the DDR2 kinase, enhancing kinase autophosphorylation rate and enzymatic activity.

## Materials and methods

### Materials

Collagen I from rat tail was from Sigma-Aldrich (C7661). BSA, Tris(2-carboxyethyl)phosphine hydrochloride (TCEP), dithiothreitol (DTT), and Tween-20 were from ThermoFisher. Restriction enzymes were from New England Biolabs (NEB). Blue prestained protein standard ladder (11–250 kDa) was from NEB. Axltide substrate peptide with the amino acid sequence of KKSRGDYMTMQIG was purchased from GenScript Biotech.

### DNA constructs

The primers used for generating the constructs are shown in S1 Table.

The full length constructs, DDR2-Y740C, DDR2-L610P, and Flag-GS-DDR2 (N-terminal Flag tag insertion with a GSGSGS linker) were generated using the QuickChange method [34], using WT DDR2 as the template. The insertion site of the Flag tag was located three amino acids after the predicted signal peptide cleavage site. All constructs were cloned into the mammalian expression vector pcDNA™3.1/Zeo(+) (ThermoFisher) for transient expression in human embryonic kidney (HEK) 293 cells.

To obtain soluble recombinant DDR2 kinases, the cDNAs encoding the DDR2 JM4 segment with the WT or mutant kinase domain (residues Val518-Glu855) were cloned into pOPINF vector (Protein Production UK), or other pOPIN vectors for expression optimisation (see S2 Table), using In-Fusion cloning (Takara). The pOPINF vector contains sequences encoding an N-terminal His$_6$-tag followed by a human rhinovirus 3C cleavage site. The pOPIN vectors used for optimisation of DDR2-L610P protein expression contain additional fusion tags before the 3C cleavage site to increase protein stability or solubility (see S2 Table). The full-length DDR2 constructs in pcDNA vector were used as PCR templates for cDNA amplification. The PCR products were incubated with pOPIN vector linearised with *Kpn*I and *Hind*III, together with 1X In-Fusion HD enzyme premix at 50°C for 15 minutes. The mixture was then transformed into Stellar™ competent cells for plasmid amplification, followed by DNA isolation and purification.

All DNA constructs were verified by Sanger Sequencing (GENEWIZ, Azenta).

## Mammalian cell culture

HEK293 and Cos-7 monkey fibroblast-like kidney cells were obtained from American Tissue Culture Collection and cultured in Dulbecco's modified Eagle medium/nutrient mixture F-12 (DMEM/F-12; Gibco, ThermoFisher) supplemented with 10% (v/v) heat inactivated foetal bovine serum (Gibco, ThermoFisher), 2 mM L-glutamine (Gibco, ThermoFisher), 100 U/mL penicillin and 100 μg/mL streptomycin (Gibco, ThermoFisher). Cells were grown at 37°C with 5% $CO_2$, 100% humidity.

## Collagen-induced DDR2 autophosphorylation

This assay was performed as described before [35]. In brief, HEK293 cells were transfected with full-length WT or mutant DDR2 constructs using polyethylenimine (PEI) transfection. The optimal ratio was 3:1 of PEI to DNA (w/w). After 24 hours of transfection, cells were serum starved for 16 hours before being stimulated with 10 μg/mL collagen I or left unstimulated for 90 minutes at 37°C. After stimulation, cells were lysed in lysis buffer (50 mM NaCl, 50 mM Tris pH 7.4, 1 mM EDTA, 1% (v/v) IGEPAL CA-630), with 1X cOmplete EDTA-free protease inhibitor cocktail (Roche), 5 mM NaF, and 1 mM sodium orthovanadate immediately added before use. Lysate aliquots were boiled at 100°C for 5 minutes before being analysed by SDS-PAGE on a 7.5% polyacrylamide gel and analysed by Western blotting.

## Flow cytometry

HEK293 cells were transfected with full-length WT or mutant Flag-GS-DDR2 constructs. Cells were collected with enzyme-free cell dissociation solution (Sigma-Aldrich) after two days of transfection and resuspended in PBS containing 1% BSA (w/v). Cells were then incubated with mouse anti-Flag M2 antibody (diluted 1:200 in PBS/BSA; Sigma-Aldrich; F3165; lot SLCC4005) on ice for 30 minutes, followed by three washes in PBS/BSA and staining with anti-mouse IgG-FITC (diluted 1:500 in PBS/BSA; Sigma-Aldrich; F9006) on ice for 30 minutes in the dark. Cells were washed once in PBS/BSA and once in PBS before being incubated with Zombie NIR™ fixable viability dye (diluted 1:1,000 in PBS; Biolegend) on ice for 20 minutes in the dark. Cells were washed twice in PBS/BSA and then fixed in PBS containing 2% formaldehyde (v/v). Data were collected on a BD Accuri C6 Plus (BD Life Sciences) using BD Accuri C6 Plus software and further analysed by FlowJo software (BD Life Sciences).

## Production of soluble recombinant DDR2 kinase constructs

The production of soluble DDR2 JM4-kinase constructs (termed DDR2-K in this manuscript) using a baculovirus expression system followed the same protocol as described before for soluble DDR1 kinase production [29]. Recombinant baculovirus was generated by co-transfecting *Spodoptera frugiperda* 9 (Sf9) cells grown at 1 x 10$^6$ cells/well in a 6-well plate with 1 μg of pOPINF-DDR2 constructs (S2 Table) and 1 μg of *Bsu36*I-linearised ORF1629-deficient bacmid (provided by Ian Jones, University of Reading). The cells were cultured in Insect Xpress medium (Lonza) without antibiotics, and Cellfectin II (ThermoFisher) was used for transfection according to the manufacturer's protocol. After culturing at 27°C for seven days, supernatant (virus) was harvested to infect Sf9 cells grown at 1 x 10$^6$ cells/mL in Insect Xpress medium, followed by culturing in an INFORS HT Multitron shaker at 27°C, 120 rpm for seven days. This virus amplification step was repeated once before protein production.

For recombinant protein production, Sf9 cells (0.5–1 L at 1 x 10$^6$ cells/mL) were infected with virus and cultured in an INFORS HT Multitron shaker at 27°C, 120 rpm for three days. Cells were pelleted and frozen at −80°C for at least one hour before cell lysis. The cell pellet was resuspended in 500 mM NaCl, 50 mM Tris pH 7.5, 0.2% Tween-20, with 0.05 U/mL benzonase (Sigma-Aldrich) and cOmplete EDTA-free protease inhibitor cocktail tablets before being sonicated on ice for 10 minutes (9 seconds on and 9 seconds off, 50% amplitude). The sonicated lysate was centrifuged at 30,000 xg for

 

one hour at 4°C, followed by filtration through a 0.45 µm cellulose acetate filter (Sartorius). The clear lysate was loaded onto a 1 mL HisTrap HP column (Cytiva) using an ÄKTA pure chromatography system (Cytiva). The protein was eluted in 500 mM NaCl, 50 mM Tris pH 7.5, 500 mM imidazole. To remove the His$_6$-tag, 1 U HRV 3C protease (ThermoFisher) was added per 0.1 mg of purified protein. The mixture was transferred to 3,500 MWCO SnakeSkin™ dialysis tubing (ThermoFisher) and dialysed in solution containing 50 mM Tris pH 8.0, 200 mM NaCl, 0.5 mM TCEP at 4°C overnight. The dialysed protein solution was loaded onto the same HisTrap HP column with the protein lacking the His$_6$-tag collected in the flow-through, followed by concentrating to 500 µL with a 10,000 MWCO Vivaspin centrifugal concentrator (Sartorius). Final purification was performed by size exclusion chromatography on a Superdex™200 Increase 10/300 GL column (Cytiva). 200 mM NaCl, 25 mM HEPES pH 7.5, 0.5 mM TCEP was used as the running buffer. Protein fractions were concentrated before snap-freezing in liquid nitrogen. The final yields for WT and Y740C kinase proteins were 2.7 mg and 1 mg per litre of Sf9 culture, respectively.

Construct DDR2-K-L610P failed to produce soluble protein with the above expression system. In an attempt to increase protein solubility, we used pOPIN constructs encoding DDR2-K-L610P with various fusion tags in Sf9 cells (S2 Table). In addition, the Sf9 lysis buffer was modified by varying the pH from 7.0 to 8.0, addition of 10% (v/v) glycerol or 2% (v/v) Tween-20. Unfortunately, none of the protocol modifications resulted in the generation of soluble DDR2-L610P JM4-kinase protein from the baculovirus system.

To check whether soluble DDR2-K-L610P protein could be detected in the cytosol of HEK293 cells, we transfected the pOPIN constructs encoding WT or mutant DDR2 JM4-kinase (S2 Table) using PEI transfection. Two days after transfection, cells were lysed at 4°C on a spin rotor. The lysis buffer contained 50 mM NaH$_2$PO$_4$ pH 8.0, 300 mM NaCl, 10 mM imidazole, 1% (v/v) IGEPAL CA-630, with 1X cOmplete protease inhibitor, 5 mM NaF, and 1 mM sodium orthovanadate immediately added before use. Soluble DDR2 proteins were then isolated from the lysates using the cobalt-based immobilised metal affinity chromatography (IMAC) Dynabeads™ (ThermoFisher) according to the manufacturer's instructions. The beads/protein mixtures were then incubated in 40 µL kinase buffer I (10 mM MnCl$_2$, 10 mM MgCl$_2$, 200 mM NaCl, 25 mM Tris pH 7.5, 100 µM DTT, 0.1 mg/mL BSA) with or without 1 mM ATP at 20°C for 30 minutes before boiling with reducing sample buffer. The samples were then analysed on 10% SDS-PAGE followed by Western blotting.

## Recombinant kinase protein autophosphorylation

The soluble DDR1-WT JM4-kinase protein (termed DDR1-K-WT) was generated and purified as described in our previous study [29]. Soluble DDR1 and DDR2 JM4-kinase proteins at 0.1 and 1 µM concentration were incubated in kinase buffer I with or without 1 mM ATP for up to 60 minutes at 20°C. For the 180 minutes time course assay, 100 µM DDR1-K-WT and DDR2-K-WT were incubated with 20 mM ATP in kinase buffer II (20 mM MnCl$_2$, 20 mM MgCl$_2$, 200 mM NaCl, 25 mM Tris pH 7.5, 100 µM DTT). The reactions were either terminated by boiling samples with reducing sample buffer followed by analysis with 10% SDS-PAGE and Western blotting, or by mixing with 80 mM EDTA solution and non-reducing sample buffer and analysis on 7.5% native-PAGE followed by InstantBlue Coomassie (Abcam) staining.

To generate the fully phosphorylated JM4-kinase constructs (DDR1-K-WT-FP and DDR2-K-WT-FP), 100 µM DDR1-K-WT and DDR2-K-WT, respectively, were incubated with 20 mM ATP in kinase buffer II for 4 hours at 23°C, followed by buffer exchange using a Vivaspin 500 concentrator (Cytiva) into 200 mM NaCl, 20 mM HEPES pH 7.5, 1 mM TCEP. Aliquots were snap-frozen in liquid nitrogen.

## Western blotting

Proteins were transferred onto 0.2 µm nitrocellulose membrane (Cytiva) and the membranes were blocked with Intercept (PBS) Blocking Buffer (LI-COR). The membranes were then incubated with primary antibody prepared in Intercept blocking buffer with 0.2% (v/v) Tween-20 overnight at 4°C. Secondary antibodies were diluted in PBS containing

0.1% Tween-20 and the incubation was performed at room temperature for 1 hour. The membranes were washed with PBS containing 0.1% Tween-20 after each antibody incubation. The blot images were obtained on an Odyssey Fc imager (LI-COR) with the image preparation and intensity quantification performed using ImageStudio Lite software (LI-COR).

The primary antibodies for determining DDR1 or DDR2 phosphorylation levels were mouse monoclonal anti-phosphotyrosine (anti-pY), clone 4G10 (1:666; Sigma-Aldrich; 05−321; lot 3015310), rabbit polyclonal anti-pY JM4 #1 (1:666; custom-made by Biomatik), rabbit polyclonal anti-pY JM4 #2 (1:666; custom-made by Biomatik), rabbit monoclonal anti-pY-DDR1/DDR2 (anti-pY A-loop; 1:666; Bio-Techne; MAB25382; lot CJCT0218092). The custom-made antibodies were validated for specificity against the JM4 segment phospho-tyrosines and described previously [29]. Anti-pY JM4 #1 binds pY569 in DDR1 and pY521 in DDR2; anti-pY JM4 #2 binds to pY586 in DDR1 and pY538 in DDR2. Mouse mono-clonal anti-DDR1, C-terminal epitope (1:500; Santa-Cruz; sc-374618; lot A2422), goat polyclonal anti-DDR2, N-terminal epitope (1:666; Bio-Techne; AF2538; lot UUA011), and rabbit polyclonal anti-DDR2, C-terminal epitope (1:1,000; Invitrogen; PA5−79144; lot VC2964102A) were used to determine total DDR1 or DDR2 expression levels. The secondary antibodies were IRDye 680RD goat anti-mouse IgG (1:17,500; LI-COR; 926−68070; lot D20316-15), IRDye 680RD donkey anti-goat IgG (1:17,500; LI-COR; 926−68074; lot C80207-04), and IRDye 800CW donkey anti-rabbit IgG (1:17,500; LI-COR; 926−32213; lot D01216-10).

## Differential scanning fluorimetry

The purified soluble DDR1-K-WT-FP protein was generated as described previously [29]. To assess thermal stability, we used differential scanning fluorimetry in a similar manner as in our previous work [29]. To this end, soluble DDR1 and DDR2 JM4-kinase proteins at 10 µM concentration were analysed with SYPRO Orange (1:1,000; ThermoFisher) in buffer containing 25 mM HEPES pH 7.5, 200 mM NaCl, 0.5 mM TCEP. The assay volume was 20 µL and the reactions were performed in a MicroAmp® Fast 96-well reaction plate (ThermoFisher). Proteins were heated from 24 to 94°C in a 7500 Fast Real-time PCR system (ThermoFisher) using 7500 Fast software. The data were analysed by Protein Thermal Shift software (ThermoFisher) using the Boltzmann equation and GraphPad Prism 9 for figure generation.

## Protein kinase kinetic analysis

The catalytic activity of soluble DDR1-K and DDR2-K proteins was analysed using the ADP-Glo™ assay (Promega) or ADP-Glo™ Max assay (Promega). The reactions were done in 25 µL volumes, where 0.1 µM soluble recombinant DDR1 or DDR2 protein was independently incubated with a range of Axltide peptide concentrations (0–520 µM) in the presence of a fixed ATP concentration (1 mM), or a range of ATP concentrations (0–1667 µM) in the presence of a fixed Axltide peptide concentration (520 µM). The reactions were performed at 22°C in 96-well Cellstar® plates (Greiner) and stopped at 10, 20, and 40 minutes. Luminescence signals were measured on a Spark® plate reader (Tecan) with Spark Control software. The ADP production was linear with respect to time and enzyme concentrations up to the last time point in all reactions; initial rates at each peptide or ATP concentration were determined by linear regression (nM•min$^{-1}$) and plotted as specific activity (SA) (nM•min$^{-1}$•mg$^{-1}$). The data were fitted with the Michaelis-Menten equation using GraphPad Prism 9 software.

## Statistical analysis

Data are presented as mean with SEM. GraphPad Prism 9 was used to perform the comparison test and establish the statistical significance between groups using two-way ANOVA with Tukey's or Šidák's multiple comparisons tests. A p-value smaller than 0.05 was judged as 'significant'.

## Results

### Missense mutations p.L610P and p.Y740C result in constitutive DDR2 phosphorylation in cells

The TKDs of human DDR1 and DDR2 and their N-terminal regulatory JM4 segments are highly conserved (Fig 1A, S1 Fig). The Warburg-Cinotti syndrome-related missense variants (p.L610P and p.Y740C) affect conserved residues in the DDR2 TKD (Fig 1A, S1 Fig). We conducted a structural analysis using the experimentally determined DDR2 kinase structure (PDB code: 7AYM [36]) and *in silico* prediction of the affected amino acid residues. Leu610 is located near the conserved catalytic lysine (Lys608) of the β3 strand, which forms a salt bridge with the αC helix residue Glu625 in the active kinase conformation (Fig 1B). Inserting a structurally rigid proline at this location might affect the positioning of the αC helix relative to the rest of the kinase N-lobe and the A-loop. This reorientation could potentially stabilise ATP binding and the active kinase conformation, suggesting how constitutive kinase activity could be achieved. Moreover, in the experimentally determined JM4-DDR1 kinase structure (PDB code: 6Y23 [29]), Leu657 (equivalent to Leu610 in DDR2) forms a hydrophobic contact (~3.6 Å) with Val580 (S2A Fig), a residue within the JM4 hairpin (Leu577 to Ala587) that penetrates deeply into the kinase active site cleft [29] (S2B Fig). The proximity of these two residues suggests that their interaction may contribute to the stability of the autoinhibited conformation mediated by the A-loop. Given the fact that the JM4 segments and kinase domains are highly conserved between DDR1 and DDR2, it is possible that a similar intramolecular interaction exists in DDR2, and a mutation at the corresponding site, such as Leu610P, that disrupts this interaction may destabilise the autoinhibited conformation.

Tyr740 is the second tyrosine residue on the A-loop, contributing to kinase autoinhibition by forming hydrogen bond interactions with Asp710 and Arg714 (Fig 1B). Substitution to cysteine abolishes these hydrogen bonds, which is predicted to favour release of the A-loop. This would allow the kinase to adopt the active conformation and exhibit increased catalytic activity. Both substitutions are predicted to be damaging evolutionarily by sequence-based tools (SIFT tool: http://sift.bii.a-star.edu.sg/; CADD tool: http://cadd.gs.washington.edu/snv; SIFT score = 0, CADD score = 31 for p.L610P; SIFT score = 0, CADD score = 30 for p.Y740C).

To study the effect of Warburg-Cinotti variants on the autophosphorylation of full-length DDR2, we chose transiently transfected HEK293 cells, a well-established model to assess DDR autophosphorylation. While HEK293 cells do not recapitulate physiological conditions of endogenously expressed receptors, they are widely used to study receptor signalling, particularly the effect of mutations on receptor functions, due to their ease of transfection and ability to express a wide range of human receptors [37]. HEK293 cells do not express endogenous DDR2, as demonstrated by our previous work [35,38–41]. DDR2 is known to autophosphorylate very slowly when stimulated with collagen [17,18]; we thus expressed DDR2 constructs in HEK293 cells and stimulated the cells with collagen I for 90 minutes, as in our previous studies [35,38–41]. While WT DDR2, as expected, showed low levels of phosphorylation in the absence of collagen and markedly increased collagen-induced phosphorylation of the biosynthetically mature, upper molecular weight forms, both DDR2-L610P and DDR2-Y740C showed collagen-independent phosphorylation, with no significant increase in phosphorylation levels upon collagen incubation, indicating that both variant receptors are constitutively active (Fig 2). We note that, compared with WT DDR2, the DDR2 variants showed a much lower proportion of the two high molecular-weight forms that represent the mature (phosphorylatable) DDR2 protein while expressing similar levels of the lowest molecular weight form (Fig 2A, **lowest panel**). The lowest molecular-weight form is an immature biosynthetic precursor protein, as confirmed by its sensitivity to EndoH digestion (S3A Fig). We therefore examined whether the mutant DDR2 proteins had any trafficking issues in reaching the cell surface during biosynthesis. Due to the lack of commercial antibodies that bind to native DDR2 on the cell surface, Flag-tagged DDR2 constructs were produced and their localisation assessed by immunofluorescence microscopy and flow cytometry using an antibody against the Flag tag. Using both methods, we observed cell surface expression of mutant DDR2 constructs albeit with somewhat lower levels than that of WT DDR2 (S3B to S3D Figs), indicating that both DDR2 variants are trafficked to the cell surface, despite the fact that much smaller proportions of mature protein were observed by Western blotting.

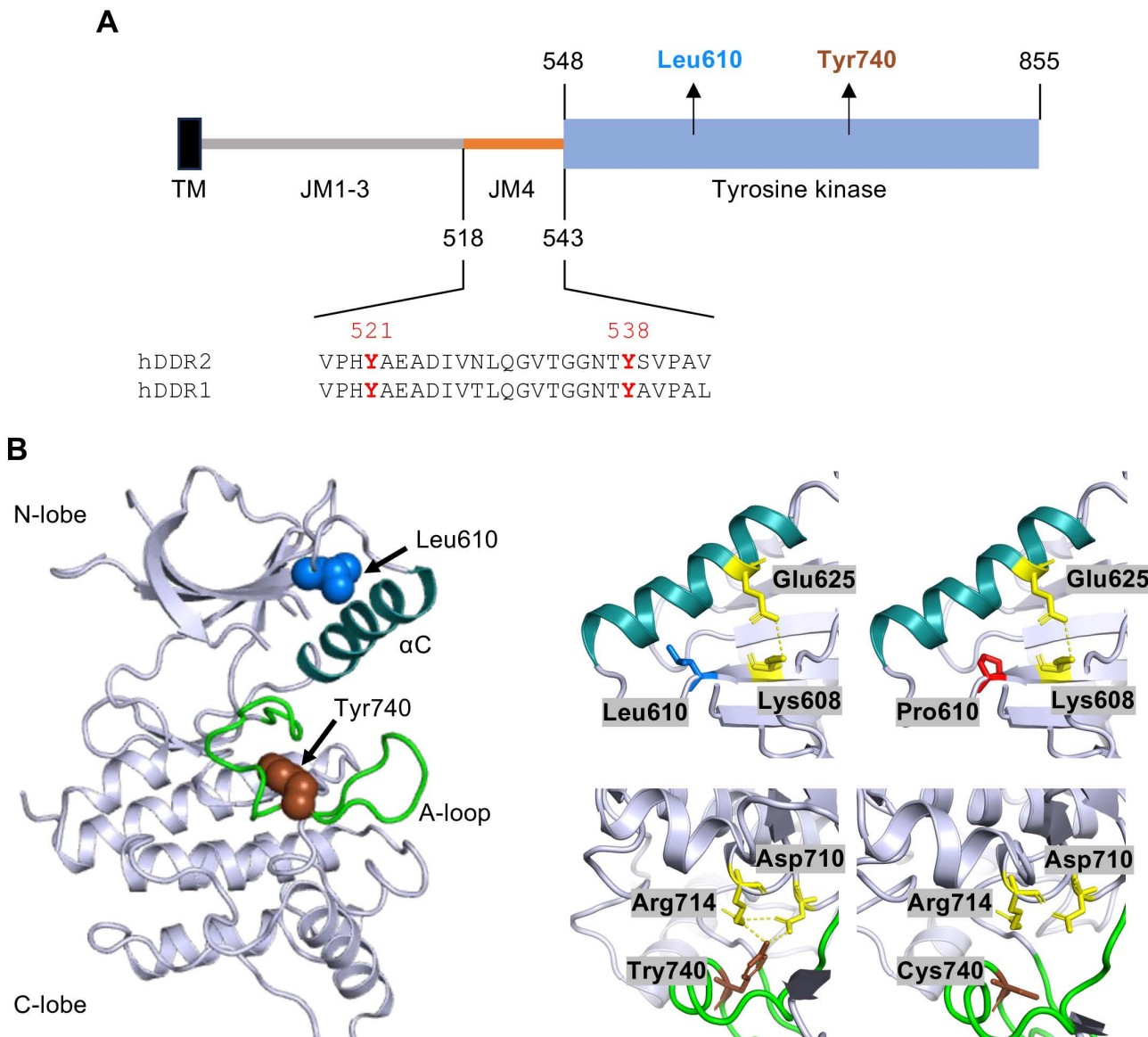

**Fig 1. Identification of Warburg-Cinotti Syndrome associated DDR2 mutations. A.** Schematic of the DDR2 cytosolic region (amino acids 400 to 855) with amino acid boundaries of the JM4 region and the kinase indicated. TM, transmembrane domain. **B. Left**, Crystal structure of DDR2 kinase domain. Figure generated with PyMOL using PDB code 7AYM [36]. The A-loop is shown in green. The αC helix is shown in cyan and labelled with 'αC'. Residues that are mutated in Warburg-Cinotti syndrome, are labelled as 'Leu610' and 'Tyr740', with Leu610 in blue located before the αC helix and Tyr740 in brown on the A-loop. **Right**, Close-up view of the residues that are mutated in Warburg-Cinotti Syndrome. Mutagenesis prediction was performed using PyMOL. Selected side chains are presented in atomic detail. **Upper left,** β3 strand residue Lys608 (in yellow) forms a salt bridge with the αC helix residue Glu625 (in yellow). Leu610 is presented in blue with the side chain positioned towards the αC helix; **Upper right,** the predicted position of Pro610 is shown in red. **Lower left**, WT Tyr740 (in brown) forms hydrogen bonds with Asp710 and Arg714 (both in yellow). The A-loop is presented in green. **Lower right,** in the Y740C variant, the hydrogen bonds with Asp710 and Arg714 are no longer present due to the substitution with Cys740. NB: this panel is rotated relative to the left-hand panel in order to visualise better the relevant interactions.

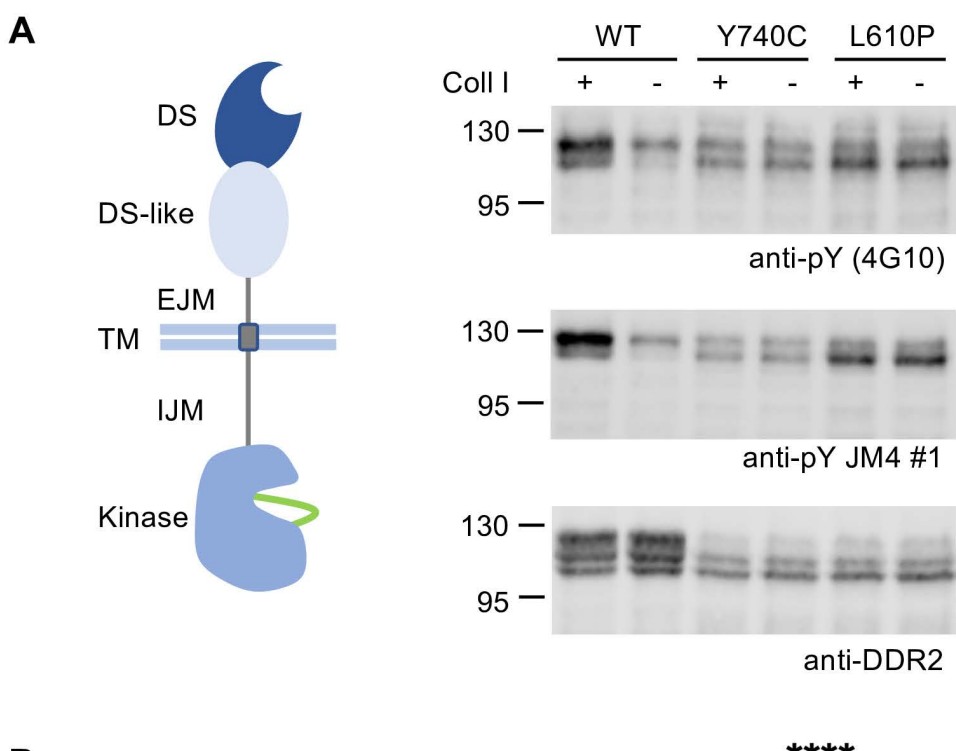

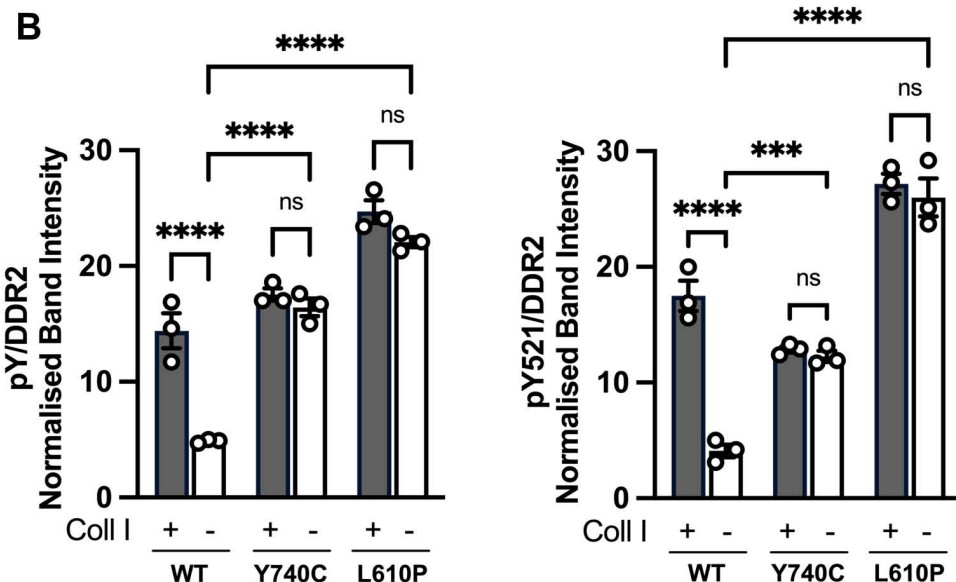

**Fig 2. Full-length DDR2-Y740C and DDR2-L610P are constitutively active in the absence of collagen stimulation. A. Left**, Schematic of the DDR2 architecture, showing the extracellular DS domain followed by a DS-like domain, an extracellular juxtamembrane (EJM) region and a TM domain. The intracellular part is composed of a flexible intracellular juxtamembrane (IJM) region and a catalytic TKD. The kinase A-loop is represented in green. **A. Right**, Expression constructs encoding full-length DDR2-WT, DDR2-Y740C, or DDR2-L610P were transfected into HEK293 cells. Cells were cultured for 24 hours, and then serum starved for at least 16 hours. Cells were then stimulated with 10 μg/mL soluble collagen I at 37°C for 90 min. Cell lysates were analysed by SDS-PAGE and Western blotting with anti-pY antibody 4G10 (binds to phospho-tyrosines independent of context) or the DDR-specific anti-pY JM4 #1 antibody (binds to pY538 in DDR2). Total DDR2 levels were then detected using an anti-DDR2 antibody. The positions of molecular weight markers (in kDa) are shown on the left. Experiments were repeated three times with similar results. **B.** Quantification of the anti-pY (4G10) antibody signals normalised to respective anti-DDR2 signals. The anti-pY signal is expressed as a percentage of the sum of all bands on a blot, with mean and standard error of the mean shown (n = 3). Statistically significant differences (P < 0.05) between signals are indicated by asterisks (**P = 0.0060; ***P = 0.0007; ****P < 0.0001; ns, non-significant). Two-way ANOVA with Tukey's multiple comparisons test.

## Soluble DDR2 WT JM4-kinase protein undergoes fast stepwise autophosphorylation

Our previous study defined the stepwise autophosphorylation of DDR1 kinase and its catalytic parameters [29]; however, this mechanism has not yet been established experimentally for DDR2. To investigate the catalytic activities of both DDR2 WT and disease variant kinases, we produced soluble DDR2 JM4-kinase (termed DDR2-K) proteins encompassing the TKD and the preceding JM4 segment (residues 518–855). Unphosphorylated WT and Y740C proteins were generated and purified from a baculovirus expression system; these constructs are referred to as DDR2-K-WT and DDR2-K-Y740C, respectively. However, the same approach did not result in soluble DDR2-K-L610P protein. We attempted to increase the solubility and stability of DDR2-K-L610P by employing different fusion tags (see S2 Table) but failed to obtain soluble DDR2-K-L610P after large-scale purification (see details in Methods). In the hope that using a mammalian cell expression system could enable us to obtain sufficient soluble DDR2-K-L610P protein for *in vitro* characterisation, we transfected HEK293 cells with the tagged DDR2-K-L610P constructs. The GST- and MBP-tagged L610P constructs showed only a limited amount of protein expression, but SUMO-tagged L610P was produced to similar amounts as the respective WT SUMO-tagged construct and used for the preliminary *in vitro* experiment described below.

To assess the ability of the SUMO-tagged WT and L610P constructs to undergo autophosphorylation *in vitro*, we isolated the recombinant proteins using Dynabeads followed by incubation with 1 mM ATP at 20°C for 30 min. His-tagged DDR2-K-WT construct (same amino acid sequences as that obtained by baculovirus expression) served as a control. JM4 and A-loop phosphorylation was detected by custom-made or commercial anti-pY specific antibodies, as in our previous study on DDR1 kinase [29]. The two WT DDR2 constructs showed a robust autophosphorylation signal, demonstrating the ability of the kinase domain to be catalytically active while still bound to the beads (S4 Fig). Although DDR2-K-L610P protein was detectable after the affinity isolation to a similar extent as DDR2-K-WT, only minimal *in vitro* autophosphorylation of the JM4 tyrosine Y521 was detected and no phosphorylation at all on the A-loop (S4 Fig). We conclude that the recombinant DDR2-K-L610P protein is unstable, leading to a loss of *in vitro* kinase function.

Our previous study showed that DDR1 kinase is released from auto-inhibition by being first phosphorylated in the JM4 segment, followed by A-loop phosphorylation [29]. Given that the JM4 segment and the TKD are highly conserved between DDR1 and DDR2, we would expect the same activation mechanism for both receptors. To confirm this, we assessed the ability of DDR1-K-WT and DDR2-K-WT proteins to autophosphorylate *in vitro*. WT DDR1 and DDR2 proteins were incubated at 1 µM in the presence of 1 mM ATP and phosphorylation detected by anti-pY specific antibodies. We observed that the two JM4 tyrosine residues in both DDR1 and DDR2 became autophosphorylated within 5 min of incubation, as detected by the anti-pY JM4 antibodies (Fig 3, **top two blots**). For DDR1, very weak A-loop phosphorylation was detected only after 30 min of incubation, as previously observed [29]. In sharp contrast, for DDR2, a clear A-loop phosphorylation signal could be detected within 5 min of incubation (Fig 3, **third blot**), suggesting faster activation of the DDR2 kinase. To further study the autophosphorylation differences, we then incubated WT DDR1 and DDR2 JM4-kinase proteins at 100 µM in the presence of a higher concentration of ATP, followed by analysis of the samples on native PAGE with Coomassie staining or denatured PAGE with Western blotting. We used the higher concentration of kinase and ATP, as the intermolecular DDR1 A-loop phosphorylation is barely visible in dilute solution (Fig 3, **third blot** and [29]). Both proteins converted to a higher mobility form within 5 min of incubation with ATP, as observed on native PAGE (Fig 4 **top panel**). In the same time period, we observed strong JM4 phosphorylation of DDR1 and DDR2, indicating correspondence between the first protein form and JM4 phosphorylation (Fig 4, **bottom panels**). Then, DDR1 gradually converted to the second and third forms over 30–60 min of incubation with ATP, matching the slow progression of A-loop phosphorylation detected by the anti-pY A-loop antibody. In contrast, DDR2 showed faster conversion, with A-loop phosphorylation being detected by the antibody after 5 min of incubation with ATP, and the conversion to the third phosphorylated form observed from 30 min of ATP incubation. Taken together, these data demonstrate that DDR2 kinase follows the same stepwise activation mechanism as DDR1 but with a higher autophosphorylation rate.

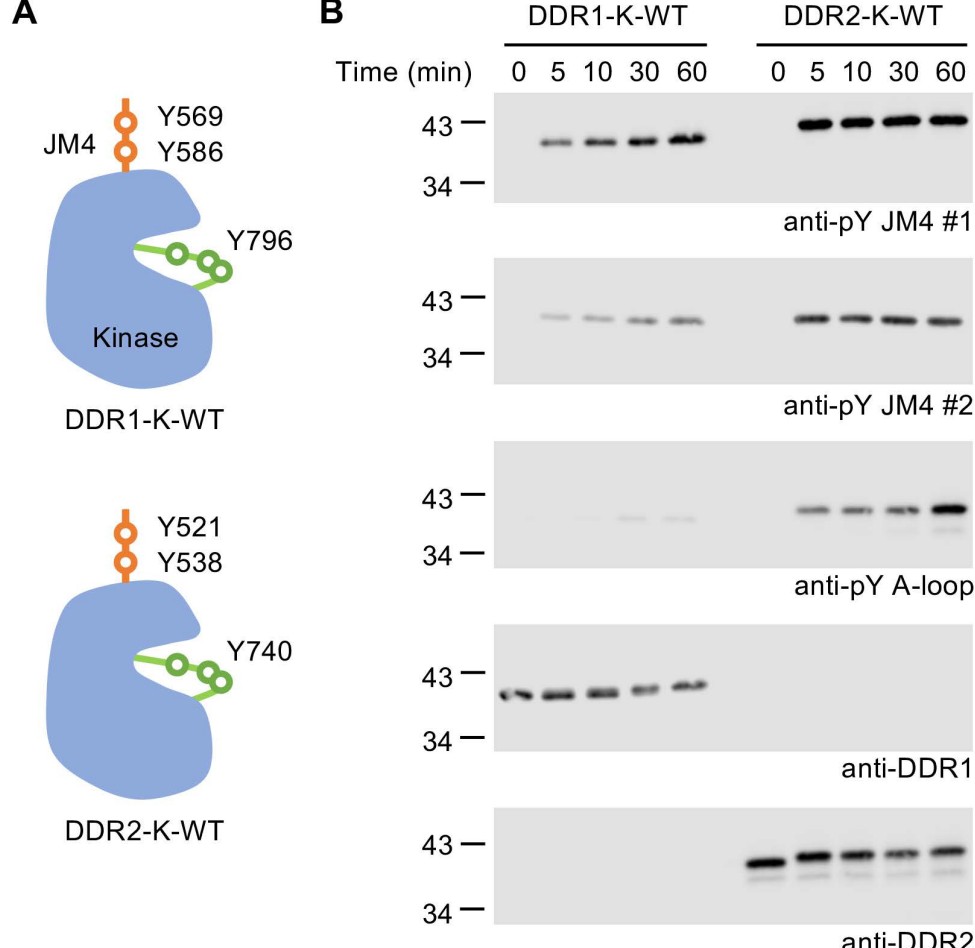

**Fig 3. Stepwise kinase activation of DDR1 and DDR2 (1 µM protein autophosphorylation assay). A.** Schematic diagrams showing the location of key JM4 and A-loop tyrosine residues in DDR1-K-WT and DDR2-K-WT constructs. Anti-pY JM4 #1 binds to pY569 in DDR1 and to pY521 in DDR2. Anti-pY JM4 #2 binds to pY586 in DDR1 and to pY538 in DDR2. Anti-pY A-loop antibody was raised against a phosphopeptide containing the human DDR2 Y740 site. This region is highly conserved between DDR1 and DDR2 and contains Y792, Y796, Y797 in DDR1 and Y734, Y740, Y741 in DDR2. **B.** The *in vitro* autophosphorylation assay was performed by incubating 1 µM protein constructs with 1 mM ATP in kinase buffer I at 20°C over a 60 min time course. The reactions were stopped at indicated time points by boiling with sample buffer. Samples were then analysed by SDS-PAGE and Western blotting with the JM4-specific and A-loop-specific anti-pY antibodies, as indicated. Total DDR levels were detected using anti-DDR1 or anti-DDR2 antibodies. The positions of molecular weight markers (in kDa) are shown on the left. Experiments were repeated three times with similar results.

### Relief of DDR2 A-loop autoinhibition by Warburg-Cinotti syndrome mutation Y740C

To investigate any potential differences in autophosphorylation rates between WT and Y740C DDR2 kinase proteins, we incubated the soluble recombinant JM4-kinase proteins at 0.1 µM in the presence of 1 mM ATP and performed a time course of incubation from 15 sec to 15 min. Autophosphorylation signals for the JM4 and A-loop of both proteins were detected already after 15 sec of ATP incubation (Fig 5A), with the DDR2-K-Y740C construct showing significantly stronger signals at earlier time points (Fig 5B). We note that the DDR2-K-Y740C A-loop phospho-signal plateaued after 2 min, whereas the WT signal increased steadily over the time course (Fig 5B), suggesting that DDR2-K-Y740C achieved complete A-loop activation within 2 minutes. The Y740C construct showed a lower A-loop phospho-signal at 5 min of ATP incubation compared with that of the WT construct, which is likely due to the mutant containing two instead of three phosphorylatable tyrosine residues in its A-loop. (The epitope of the anti-pY A-loop antibody contains all three A-loop tyrosine

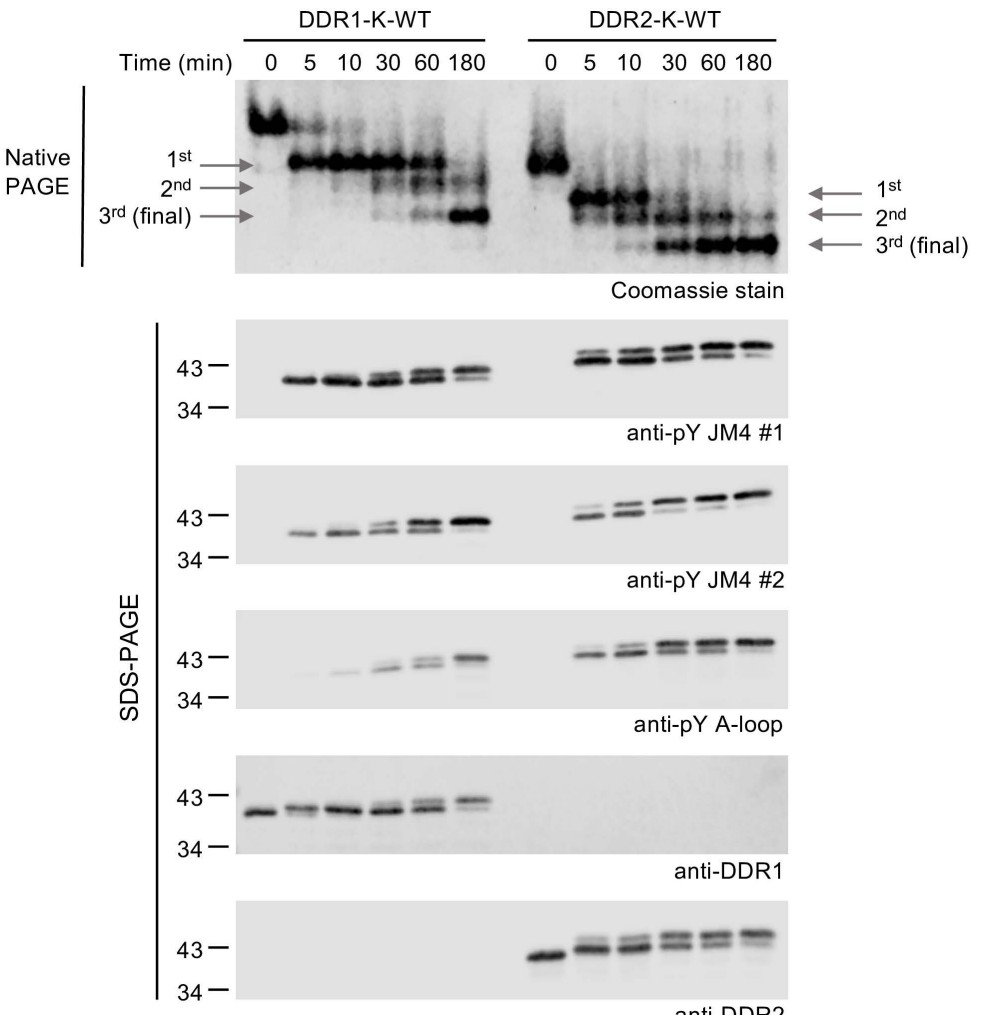

**Fig 4. Stepwise kinase activation of DDR1 and DDR2 (100 μM protein autophosphorylation assay).** The *in vitro* autophosphorylation assay was performed by incubating 100 μM DDR1-K-WT or DDR2-K-WT constructs with 20 mM ATP in kinase buffer II at 20°C over a 180 min time course. The reactions were stopped at indicated time points by adding 80 mM EDTA (for native-PAGE) or boiling with sample buffer (for SDS-PAGE). Samples were either separated by native-PAGE with Coomassie staining (upper image) or analysed by SDS-PAGE and Western blotting with the JM4-specific and A-loop-specific anti-pY antibodies (lower image). Differently phosphorylated DDR forms are indicated with grey arrows. Total DDR levels were detected using anti-DDR1 and anti-DDR2 antibodies. The positions of molecular weight markers (in kDa) are shown on the left. The experiment was performed once.

residues.) In thermal unfolding experiments, the Y740C mutation greatly reduced DDR2 kinase stability, with the mutant protein's melting temperature comparable to the previously purified [29] completely phosphorylated DDR1 kinase (termed DDR1-K-WT-FP) (Fig 5C). The DDR1-K-WT construct displayed higher thermal stability than the DDR2-K-WT construct. Taken together, our results indicate that the Warburg-Cinotti syndrome Y740C mutation disrupts the autoinhibited structure of the DDR2 kinase, thereby reducing its thermal stability and increasing the rate of autophosphorylation.

## A-loop Y740C mutation increases the enzymatic activity of DDR2 kinase

To determine the catalytic activities of unphosphorylated and fully phosphorylated (FP) DDR1 and DDR2 kinases, we generated DDR1-K-WT-FP and DDR2-K-WT-FP by incubating the purified DDR-K-WT proteins with 20 mM ATP at 23°C

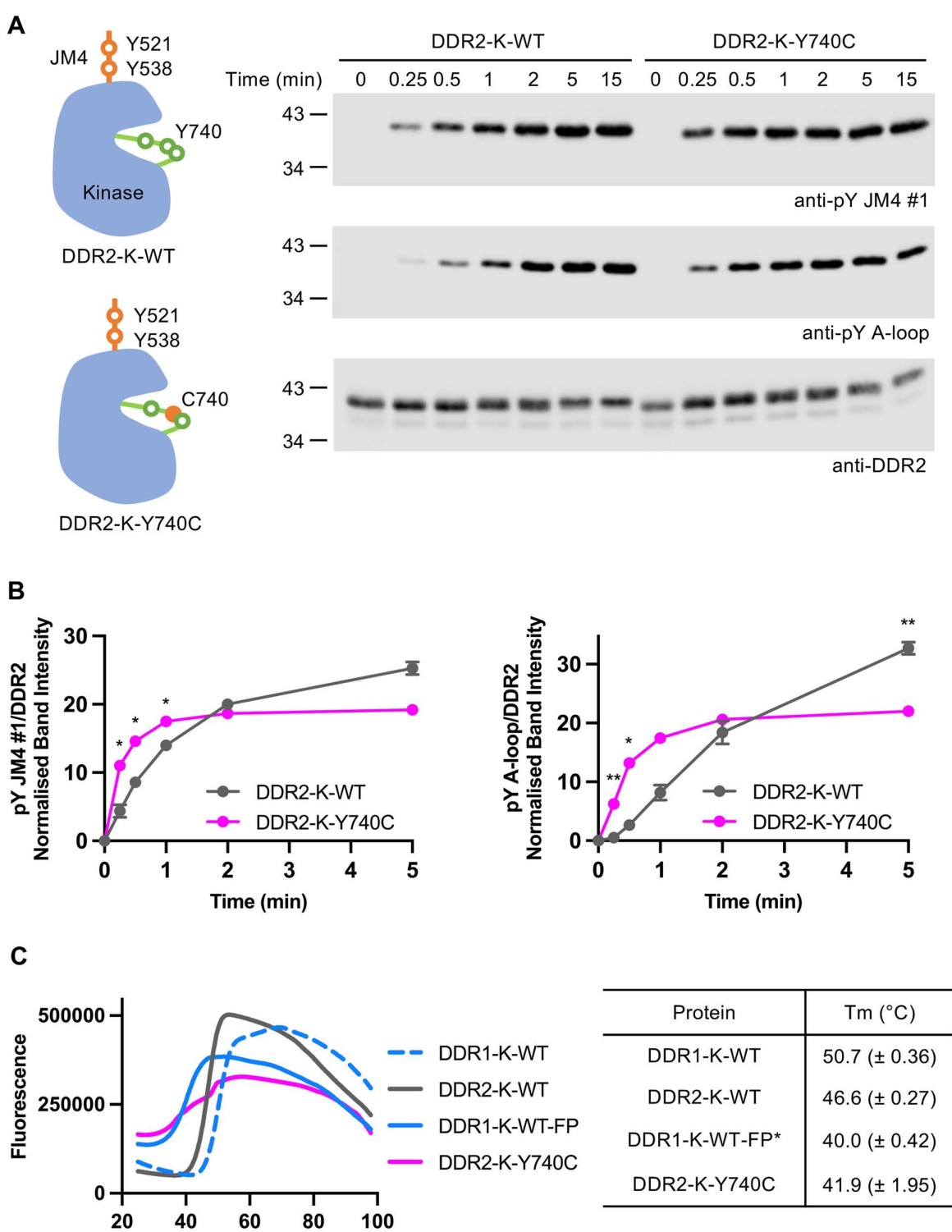

**Fig 5. DDR2-Y740C kinase is more active than DDR2-WT kinase. A.** Schematic structure of the recombinant DDR2 constructs. JM4 tyrosines (Y521, Y538) and A-loop tyrosines (replaced by C740 in DDR2-K-Y740C) are labelled. The soluble DDR2-K-WT and DDR2-K-Y740C proteins were incubated at 0.1 μM in the presence of kinase buffer I and 1 mM ATP for the indicated time course (0-15 min) at 20°C. The samples were then boiled in sample

buffer and analysed by SDS-PAGE and Western blotting. The positions of molecular weight markers are shown in kDa on the left side of each blot. Experiments were repeated three times with similar results. **B.** Quantification of the anti-pY JM4 #1 and anti-pY A-loop, signals normalised to respective anti-DDR2 signals. The anti-pY signal is expressed as a percentage of the sum of all bands on a blot, with mean and standard error of the mean shown (n = 3). Statistically significant differences (P < 0.05) between the DDR2-K-WT and DDR2-K-Y740C signals are indicated by asterisks (left graph, *P = 0.0389, t = 0.25 min; *P = 0.0270, t = 0.5 min; *P = 0.0400, t = 1 min. Right graph, **P = 0.0067, t = 0.25 min; *P = 0.0170, t = 0.5 min; **P = 0.0099, t = 5 min). Two-way ANOVA with Šídák's multiple comparisons test. **C.** The soluble DDR1 and DDR2 kinase constructs, DDR1-K-WT, DDR2-K-WT, DDR1-K-WT-FP, and DDR2-K-Y740C were analysed by differential scanning fluorimetry. Proteins at 10 µM concentration were heated with SYPRO Orange from 24 to 94°C with the fluorescence detected. The average fluorescence from three independent experiments is shown. The data were analysed by Protein Thermal Shift software (ThermoFisher) using Boltzmann equation to calculate the denaturation midpoint. GraphPad Prism 9 was used for figure generation. *The DDR1-K-WT-FP protein is the purified fully phosphorylated protein as generated previously [29].

for 4 hours. At this timepoint, the majority of the proteins migrated as their FP forms on a native gel (Fig 6A, **top panel**), with clear JM4 and A-loop phosphorylation confirmed by Western blotting (Fig 6A, **bottom panels**). Therefore, we did not purify the FP form further by ion exchange chromatography as in our previous study [29]. We then analysed the catalytic activities of the WT DDR1 and DDR2 JM4-kinase proteins in their unphosphorylated and FP states, as well as unphosphorylated DDR2-K-Y740C, using Axltide peptide as a peptide substrate. DDR1-K-WT-FP and DDR2-K-WT-FP, as well as DDR2-K-Y740C, showed Michaelis-Menten behaviour against both peptide and ATP substrates (Fig 6B). The DDR2-K-WT showed no detectable saturation with respect to both peptide and ATP substrate over the concentration ranges used (Fig 6B). The DDR1-K-WT showed substantially lower activities against both substrates, in agreement with the previous study [29]. As the unpurified DDR1-K-WT-FP used here is a mixture comprising proteins with different levels of phosphorylation, it displayed lower specific activity towards both substrates than we previously determined with purified DDR1-K-WT-FP [29]. We found much higher catalytic efficiency ($k_{cat}/K_m$) for DDR2-K-WT-FP than for DDR1-K-WT-FP (Table 1). These data indicate that DDR2 kinase exhibits greater enzymatic efficiency than DDR1 kinase. The mutant unphosphorylated DDR2-K-Y740C showed a similar $k_{cat}$ value and a slightly higher $K_m$ value for the peptide substrate to that of DDR2-K-WT-FP (Table 1). Furthermore, unphosphorylated DDR2-K-Y740C showed very similar specific activity towards peptide substrate as the phosphorylated DDR2-K-WT protein (Fig 6B). These data show that the Y740C mutation substantially enhances the enzymatic activity of the unphosphorylated DDR2 kinase, which could potentially impact DDR2-regulated signalling pathways that play a role in bone growth and wound healing processes. As a result, individuals with Warburg-Cinotti syndrome may experience issues such as bone resorption, keloid formation, and delayed wound healing [3,42].

## Discussion

Warburg-Cinotti syndrome is an autosomal dominant condition caused by the DDR2 variants DDR2-L610P or DDR2-Y740C [3]. Our results demonstrate that these variants are constitutively active for autophosphorylation when expressed as full-length proteins in cells, which is in line with the original study's proposal that the disease variants would exhibit ligand-independent receptor activation [3]. While we were not able to generate soluble kinase constructs for the L610P variant for *in vitro* characterisation, enzymatic characterisation of the soluble unphosphorylated DDR2-Y740C variant demonstrated enhanced enzymatic activity, similar to that of the respective fully phosphorylated WT construct, which supports the notion that the Warburg-Cinotti variant with the Y740C A-loop substitution predominantly adopts the catalytically active conformation of the DDR2 TKD.

An earlier study also showed that tyrosine-740 substitution in DDR2 can lead to constitutive activation. Yang et al substituted tyrosine-740 with phenylalanine, which led to enhanced catalytic activity of a construct encompassing the entire DDR2 cytosolic region [43]. While this study did not analyse DDR2 autophosphorylation in cells, cellular expression of full-length DDR2-Y740F resulted in ligand-independent constitutive activity as measured by enhanced downstream expression of matrix metalloproteinase 2 (MMP2). This study further concluded that tyrosine-740 is the only tyrosine in the DDR2 A-loop required for TKD autoinhibition [43], in line with the present

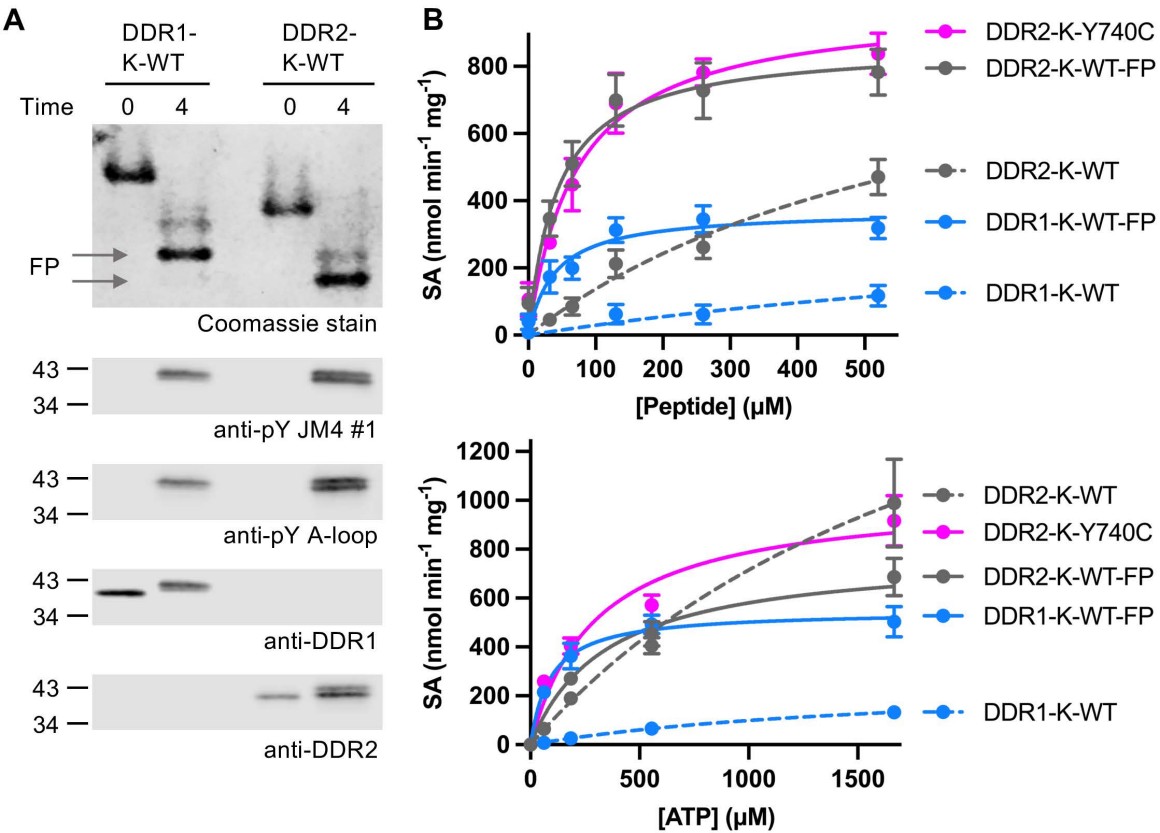

**Fig 6. A-loop cysteine mutation increases the enzymatic activity of DDR2 kinase. A.** DDR1 and DDR2 FP forms were obtained by incubating 100 µM DDR1-K-WT or DDR2-K-WT protein constructs with 20 mM ATP in kinase buffer II at 23°C for 4 hours. The reactions were stopped by adding 80 mM EDTA. Sample aliquots were either separated by native-PAGE with Coomassie staining (upper image) or analysed by SDS-PAGE and Western blotting with the JM4-specific and A-loop-specific anti-pY antibodies (lower image). DDR FP forms are indicated with grey arrows. Total DDR levels were detected using anti-DDR1 and anti-DDR2 antibodies. The positions of molecular weight markers (in kDa) are shown on the left. **B.** *In vitro* kinase activity of soluble DDR1 and DDR2 constructs was assessed using the ADP-Glo™ and ADP-Glo™ Max assays. The reactions were performed at 22°C and stopped at 10, 20, and 40 minutes to determine initial velocities. **Top**, assay performed with a range of Axltide peptide concentrations (0-520 µM) in the presence of a fixed ATP concentration (1 mM). **Bottom**, assay performed over a range of ATP concentrations (0-1667 µM) in the presence of a fixed Axltide peptide concentration (520 µM). Data were then fitted with the Michaelis-Menten equation. Mean and standard error of the mean shown (n = 3). SA, specific activity.

study. However, our results differ from the results obtained by Yang et al with regard to the requirement of Src for initial phosphorylation of DDR2: Yang et al found that in vitro activity of recombinant DDR2 kinase required the phosphorylation of DDR2 by Src [43], while we report catalytic activity without the addition of Src. A possible explanation for this discrepancy is the use of different recombinant proteins in the two studies. The former study used GST-tagged DDR2 constructs encompassing the entire DDR2 cytosolic region (reported as amino acids 441–815), while our construct encompasses only the JM4 segment and kinase domain (untagged construct with amino acid boundaries residues 518–855). Another study using the entire cytosolic region of DDR2 (His-tagged construct, amino acids 422–855) also found only minimal DDR2 catalytic activity in the absence of Src [44]. It is therefore possible that the amino acid region preceding the JM4 segment functions as an inhibitory region in the in vitro setting used in these studies. Of note, the predicted Src binding sites are contained in the JM4 region [45], which is present in our constructs. Nevertheless, cellular studies are required to understand the possible role of Src-mediated DDR2 activation in a physiologically relevant context.

**Table 1. Apparent $K_m$, $v_{max}$ and $k_{cat}$, values for Axltide peptide and ATP.**

| Substrate | Kinetic parameter | DDR1-K-WT | DDR1-K-WT-FP | DDR2-K-WT | DDR2-K-WT-FP | DDR2-K-Y740C |
|---|---|---|---|---|---|---|
| Peptide | $K_m$ (µM) | ND | 39.3 (14.3 to 84.5) | ND | 43.4 (21.2 to 79.4) | 71.6 (40.3 to 122.6 |
| | $v_{max}$ (µmol*L$^{-1}$*min$^{-1}$) | ND | 1.4 | ND | 3.3 | 3.8 |
| | $k_{cat}$ (min$^{-1}$) | ND | 14.4 (11.7 to 17.7) | ND | 33.4 (28.4 to 39.5) | 38.0 (32.2 to 45.4) |
| ATP | $K_m$ (µM) | ND | 91.9 (46.9 to 164.8) | ND | 351.5 (146.6 to 892.7) | 299.3 (142.0 to 636.0) |
| | $v_{max}$ (µmol*L$^{-1}$*min$^{-1}$) | ND | 2.1 | ND | 3.0 | 3.9 |
| | $k_{cat}$ (min$^{-1}$) | ND | 21.3 (18.4 to 24.6) | ND | 30.3 (23.0 to 43.1) | 39.5 (31.6 to 51.3) |

Kinetic parameters for the Axltide peptide and ATP were calculated by fitting initial rate data (mM min$^{-1}$) with the Michaelis-Menten equation. The mean and 95% confidence intervals are shown for 3 independent experiments. ND, not determined; NA, not available.

Gain of function mutations in RTKs are responsible for a wide range of human pathologies and developmental disorders and are often seen to arise sporadically in cancer [46–49]. While many of these mutations occur in RTK regions outside of the TKDs, A-loop mutations leading to constitutive activity have been reported for the class III RTKs [47,50]. This group of RTKs includes the KIT, FLT3 and PDGF receptor families. As for DDR2, mutations within the A-loop of type III RTKs result in constitutive activity when the affected residues are critical in stabilising the inactive conformation [50]. By releasing the inhibitory constraints, these mutations energetically favour the active kinase conformation and reduce the time the enzyme spends in the inactive form [50]. A-loop mutations leading to increased basal kinase activity have also been described for FGFR3 [51]. For instance, K650E was shown to stabilise the active A-loop conformation in FGFR3, leading to kinase activation independently of A-loop phosphorylation [52].

Our data show that, while A-loop phosphorylation is a step in the activation of WT DDR2 kinase activation, phosphorylation of A-loop tyrosine-740 is not required for DDR2 catalytic activity. This is reminiscent of the situation of c-KIT, which has a similar two-step activation mechanism as the DDRs, with a JM segment stabilising an auto-inhibitory conformation in the inactive state [53]. Activation of WT c-KIT requires JM segment phosphorylation, which is followed by A-loop phosphorylation [54]. A-loop mutation of Y823F resulted in a variant with similar catalytic activity as WT c-KIT but enhanced speed of activation for substrate phosphorylation [54]. However, mutation of A-loop tyrosines in other RTKs, such as EphA4, leads to loss of catalytic activity, with substitution with phenylalanine resulting in a decrease in substrate binding affinity [55].

Our structural prediction showed both disease variants to be damaging to the autoinhibitory conformation (Fig 1). While we were not able to obtain experimental insight into the activation mechanism of the L610P substitution, we can speculate about its effect. Leu610 is located at the end of the kinase β3 strand (Fig 1B). A proline substitution within a β sheet would be disfavoured, as it would disrupt the secondary structure and thereby increase the protein's instability [56]. The location of the proline substitution just before the αC helix, however, is predicted to affect the position of the nearby αC helix due to the conformational rigidity of proline [57]. The overall effect would be to affect the position of the αC helix, resulting in stabilising the active kinase conformation and facilitating ATP binding.

We were unable to obtain a stable form of the L610P variant as a soluble kinase construct, despite several attempts to use fusion tags designed to enhance folding of recombinant proteins. When expressed as full-length TM protein in mammalian cells, the L610P variant was found at the cell surface to similar extents as the Y740C variant (S3C **and** S3D Figs), indicating that regions within the ECD or the TM region of DDR2 are key in folding and stabilising this disease variant. It is also conceivable that tethering the kinase to a membrane prevents protein aggregation. However, both L610P and Y740C

variants showed substantially decreased proportions of the complex glycosylated forms in mammalian cells and were present at the cell surface to somewhat lower extents than WT DDR2 (Figs 2, S3A **and** S3D), indicating altered cellular trafficking of these variants and potentially different cellular localisation. This is reminiscent of the situation of activating FGFR3 mutations, where A-loop mutants, particularly those associated with severe disease, were found to reside in the endoplasmic reticulum as immature biosynthetic precursors that failed to be degraded [51,58]. Whether receptor trafficking and associated signalling from potentially different intracellular compartments plays a role in Warburg-Cinotti disease will need to be addressed in future studies.

The study by Xu et al demonstrated the L610P and Y740C variants of DDR2 to be responsible for Warburg-Cinotti syndrome [3], and our results shed light on the molecular mechanism. How increased DDR2 kinase activity leads to the clinical symptoms of the syndrome will need to await further functional studies in patient-derived cellular systems, but we can speculate that enhanced DDR2 activity induces collagen-independent MMP14 (MT1-MMP) activation in fibroblasts, which may play a key role in keloid formation. This is based on previous studies showing that DDR2 drives collagen-induced MMP14 activation in fibroblasts [59] and that enhanced MMP14 activity in keloid fibroblasts is responsible for continued collagen deposition and keloid progression [60].

## Supporting information

**S1 Table. Primer list.** Mutagenic bases are shown in bold. Flag-tag insertion sequences are shown in bold and underlined.
(DOCX)

**S2 Table. List of pOPIN constructs and their protein constructs names.** GST, glutathione S-transferase; SUMO, small ubiquitin-like modifier; MBP, maltose-binding protein.
(DOCX)

**S1 Fig. Sequence alignments of the JM4 region and TKD of DDR1 and DDR2.** Clustal Omega alignment of partial sequences of the DDR1 (amino acids 536–913) and DDR2 (amino acids 496–855) cytosolic regions. The JM4 regions are coloured in orange, the αC helices are indicated in teal and the A-loops in green. The catalytically important Lys on the β3 strand (L655 in DDR1, L608 in DDR2) and Glu in the αC helix (E572 in DDR1, E625 in DDR2) are indicated in red, and the A-loop middle Tyr (Y796 in DDR1, Y740 in DDR2) are indicated in yellow. The DDR2 residues mutated in Warburg-Cinotti disease (L610 and Y740) are shown by yellow letters highlighted in black. Sequence numbers are shown to the right. '*' (asterisk), fully conserved residue; ':' (colon), conserved substitutions; '.' (period), less well conserved substitution.
(DOCX)

**S2 Fig. DDR1-Leu657 (equivalent to Leu610 in DDR2) forms a contact with the JM4 hairpin residue. A.** DDR1 Leu657 (in blue) (equivalent to Leu610 in DDR2) forms a hydrophobic contact (~3.6 Å) with JM4 (in orange) residue Val580 (in yellow). Selected side chains are presented in atomic detail. **B.** Close-up view of the JM4 hairpin region (residue 577–587). The colour scheme is the same as in **A**. Selected side chains are presented in atomic detail.
(DOCX)

**S3 Fig. Cell surface expression of Warburg-Cinotti mutants. A**. Full-length DDR2-WT, DDR2-Y740C, DDR2-L610P were transiently expressed in HEK293 cells. The cells were lysed two days after transfection. The lysates were treated with EndoH (H) or left untreated (-), followed by boiling in sample buffer and analysis by Western Blotting with anti-DDR2 antibody. The positions of the EndoH sensitive, immature biosynthetic precursor forms are indicated by an orange arrow. The positions of the Endo H resistant mature glycoforms are indicated in blue. The positions of molecular weight markers (in kDa) are shown on the left. **B**. Expression constructs encoding Flag-DDR2-WT, Flag-DDR2-Y740C or Flag-DDR2-L610P were transfected into Cos-7 cells. Cells were subsequently stained with anti-Flag antibody, followed by AlexaFluor

488 anti-mouse IgG1 secondary antibody. Scale bar = 50 μm. Widefield images were acquired using an Olympus BX51 microscope with Simple-PCI acquisition software. **C**. Expression constructs encoding Flag-DDR2-WT, Flag-DDR2-Y740C or Flag-DDR2-L610P were singly transfected into HEK293 cells. Cells were subsequently stained with anti-Flag antibody, followed by FITC-labelled anti-mouse-Fc antibody and analysed by flow cytometry. The filled grey histograms represent secondary antibody only staining, while the open histograms represent anti-Flag and secondary antibody staining. **D.** Quantification of mean fluorescence intensity from the flow cytometry experiments.
(DOCX)

**S4 Fig.** *In vitro* **kinase activity of soluble recombinant DDR2 kinase constructs.** Constructs encoding His-tagged DDR2-K-WT, SUMO-tagged DDR2-K-WT, or SUMO-tagged DDR2-K-L610P were transfected into HEK293 cells. The His-tagged proteins were then isolated from the cell lysates with cobalt-based immobilized metal affinity Dynabeads. The purified protein/bead mixtures were then stimulated with 1 mM ATP in kinase buffer I for 30 min at 20°C. Samples were boiled in sample buffer and analysed by SDS-PAGE and Western blotting with the JM4-specific and A-loop-specific anti-pY antibodies, as indicated. Total DDR2 levels were detected using an anti-DDR2 antibody. The positions of molecular weight markers (in kDa) are shown on the left. CL, cell lysate; HP, affinity isolated His-tagged protein.
(DOCX)

**S1 File. Raw images.** Uncropped images of all blots shown, lanes labelled and annotated as in the respective figures.
(PDF)

## Acknowledgments

We thank Ray Owens (Protein Production UK) for providing the pOPIN vectors, Valerie Good for help with baculovirus expression, Alessia David for guiding the structure prediction of the disease variants, Doug Sammon for providing constructs and help with assays, and Erhard Hohenester for critical reading of the manuscript.

## Author contributions

**Conceptualization:** Birgit Leitinger.

**Funding acquisition:** Birgit Leitinger.

**Investigation:** Ziteng Hao.

**Project administration:** Birgit Leitinger.

**Supervision:** Birgit Leitinger.

**Visualization:** Ziteng Hao.

**Writing – original draft:** Ziteng Hao, Birgit Leitinger.

**Writing – review & editing:** Ziteng Hao, Birgit Leitinger.

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
