## [Decision Letter · Decision Letter 0]

22 Jul 2025

Dear Dr. Leitinger,

Thank you for submitting your manuscript to PLOS ONE. After careful consideration, we feel that it has merit but does not fully meet PLOS ONE’s publication criteria as it currently stands. Therefore, we invite you to submit a revised version of the manuscript that addresses the points raised during the review process.

We look forward to receiving your revised manuscript.

Kind regards,

Prasad Sanjay Dhiwar, PhD

Academic Editor

PLOS ONE

Journal Requirements:

3. To comply with PLOS ONE submissions requirements, please provide the following information in the Methods section of the manuscript and in the “Ethics Statement” field of the submission form (via “Edit Submission”):

* Please indicate whether an animal research ethics committee prospectively approved this research or granted a formal waiver of ethics approval.* Please enter the name of your Institutional Animal Care and Use Committee (IACUC) or other relevant ethics board. Also include an approval number if one was obtained.

* If anesthesia, euthanasia, or any kind of animal sacrifice is part of the study, please include briefly in your statement which substances and/or methods were applied.

For additional information about PLOS ONE submissions requirements for ethics oversight of animal work, please refer to http://journals.plos.org/plosone/s/submission-guidelines#loc-animal-research

Reviewers' comments:

Reviewer's Responses to Questions

**Comments to the Author**

1. Is the manuscript technically sound, and do the data support the conclusions?

Reviewer #1: Yes

Reviewer #2: Partly

2. Has the statistical analysis been performed appropriately and rigorously?

Reviewer #1: Yes

Reviewer #2: Yes

3. Have the authors made all data underlying the findings in their manuscript fully available?

Reviewer #1: Yes

Reviewer #2: Yes

4. Is the manuscript presented in an intelligible fashion and written in standard English?

Reviewer #1: Yes

Reviewer #2: Yes

Reviewer #1: The manuscript addresses an important topic and presents valuable data. After careful

evaluation, I believe it can be accepted for publication pending minor revisions to clarify

certain points and improve readability. The authors are encouraged to address the specific

comments provided to enhance the overall quality and clarity of the manuscript.

1. Please clearly state in the introduction that the DDR2-L610P variant could not be

biochemically characterized due to solubility problems, to set appropriate expectations

for the results section.

2. Briefly mention any limitations of using DDR1 as a reference model for DDR2, despite

structural similarity.

3. The hypothesis—that DDR2 mutations relieve autoinhibition and enhance kinase

activity—should be clearly articulated at the end of the introduction to guide the reader.

4. Briefly explain or break down long, complex sentences to improve clarity and

readability. a sahi hai kya?

5. Clarify and expand how the increased DDR2 kinase activity caused by mutations leads to

the clinical symptoms of Warburg-Cinotti syndrome to strengthen the biological

relevance.

6. Consider adding a brief discussion on the limitations of using HEK293 cells as a model

for DDR2 activity, as these cells may not fully recapitulate physiological conditions. OR

Please briefly discuss the limitations of using HEK293 cells as a model system for DDR2

activity, including possible differences from endogenous expression contexts.

Reviewer #2: The study attempts to find the mechanistic basis behind Warburg-Cinotti syndrome caused by mutations in DDR2. There are valuable additions in the manuscript but few things need to be addressed before it's accepted for publication, includingg some clarifications and resolving what seems to be contradicting statements.

The computational prediction using ELASPIC estimates a change in the stability of the L610P mutant (folding energy) of ΔΔG = -3 kcal•mol-1. The prediction cannot be verified due to the inability to purify an active construct. What do similar calculations show for Y740C mutant, which shows decreased stability? Showing the validity of the estimates with Y740C is needed to cement the predictions for L610P.

S3 Fig. B, the localization is not clear in black and white. Was the fluorescence obtained quantified in the images? The cell cytometer data in panel C imply there are differences in cell population, if not at expression level, then maybe localization, with the mutated DDR2 variants. How did the authors deduce that there is comparable cell surface expression? The discussion section mentions (line 676) differential cellular trafficking, which contradicts their results!

The ADP-Glo assay used is an endpoint assay that measures the total phosphorylation happening after time “t”, not a kinetic continuous assay, which should work to assess specific activity for the enzyme. I am not sure if it can be used accurately to measure Michaelis Menten kinetic constants. The measured activity might not represent the initial rate if the reaction has progressed beyond the linear phase of the reaction progress curve, where the rate is constant. The methods section mentions that the reaction was stopped at 10, 20, and 40 minutes, the earliest of which (10 min) far exceeds the 2 minutes that it takes the mutant and FP protein to fully phosphorylate the A loop per the WB data, hence not measuring the initial rates for sure. Thus, Table 1, results, and discussion should only copare the specific activity of the constructs.

I am not sure what Figure 4B adds that is different than Figure 3B (other than using 100-fold of the enzyme concentrations, which I cannot justify), since they both seem to compare the time course of in vitro autophosphorylation of DDR1 and DDR1. (Lines 447 and 448: Are these the same two constructs, and are these the same ones in line 456/457?). If all the same, I suggest merging the native gel panel with Figure 3 for a more concise figure.

In the discussion, the authors do a great job summarizing mutation effect in RTK, yet they need to acknowledge that because the constructs used lack some of the cytosolic components of the JM domain that were present in other studies, it’s harder to draw the conclusion that Src activation is not needed in a physiologically relevant environment.

**Do you want your identity to be public for this peer review?** For information about this choice, including consent withdrawal, please see our Privacy Policy

Reviewer #1: **Yes: ** NA

Reviewer #2: No

---

## [Author Response · Author response to Decision Letter 1]

3 Sep 2025

We thank the reviewers for their careful reading of the manuscript and constructive criticism. A point-by-point response is below.

Reviewer #1

The manuscript addresses an important topic and presents valuable data. After careful evaluation, I believe it can be accepted for publication pending minor revisions to clarify certain points and improve readability. The authors are encouraged to address the specific comments provided to enhance the overall quality and clarity of the manuscript.

1. Please clearly state in the introduction that the DDR2-L610P variant could not be biochemically characterized due to solubility problems, to set appropriate expectations for the results section.

Response: This information has now been added to the Introduction, lines 107-108.

2. Briefly mention any limitations of using DDR1 as a reference model for DDR2, despite structural similarity.

Response: This information has now been added to the Introduction, lines 90-94.

3. The hypothesis—that DDR2 mutations relieve autoinhibition and enhance kinase activity—should be clearly articulated at the end of the introduction to guide the reader.

Response: This information has now been added to the Introduction, lines 103-104.

4. Briefly explain or break down long, complex sentences to improve clarity and readability. a sahi hai kya?

Response: We have carefully reviewed the manuscript text to check for issues with clarity and readability.

5. Clarify and expand how the increased DDR2 kinase activity caused by mutations leads to the clinical symptoms of Warburg-Cinotti syndrome to strengthen the biological relevance.

Response: This information has now been added to the Discussion, lines 708 - 717.

6. Consider adding a brief discussion on the limitations of using HEK293 cells as a model for DDR2 activity, as these cells may not fully recapitulate physiological conditions. OR Please briefly discuss the limitations of using HEK293 cells as a model system for DDR2 activity, including possible differences from endogenous expression contexts.

Response: Further information about the use and limitations of HEK293 cells has been added into the manuscript, lines 373 – 376.

Reviewer #2

The study attempts to find the mechanistic basis behind Warburg-Cinotti syndrome caused by mutations in DDR2. There are valuable additions in the manuscript but few things need to be addressed before it's accepted for publication, includingg some clarifications and resolving what seems to be contradicting statements.

The computational prediction using ELASPIC estimates a change in the stability of the L610P mutant (folding energy) of ΔΔG = -3 kcal•mol-1. The prediction cannot be verified due to the inability to purify an active construct. What do similar calculations show for Y740C mutant, which shows decreased stability? Showing the validity of the estimates with Y740C is needed to cement the predictions for L610P.

Response: We have now used the ELASPIC server to calculate the folding energies for both mutants. Frustratingly, the server now gives positive ΔΔG values for both proteins. Using a different server (DDGEMB) gives ΔΔG = -1.33 kcal•mol-1 for L610P, which is defined as destabilizing, and ΔΔG = -0.32 kcal•mol-1 for Y740C, which is defined as neutral. While the latter calculations are in line with our inability to produce the L610P construct while the Y740C could be obtained as a folded protein, we do not believe these calculations to be reliable and decided to remove the prediction of the folding energy of the L610P mutant and only describe experimentally obtained data in the manuscript with regard to protein stability.

S3 Fig. B, the localization is not clear in black and white. Was the fluorescence obtained quantified in the images? The cell cytometer data in panel C imply there are differences in cell population, if not at expression level, then maybe localization, with the mutated DDR2 variants. How did the authors deduce that there is comparable cell surface expression? The discussion section mentions (line 676) differential cellular trafficking, which contradicts their results!

Response: We now show quantification of the flow cytometry data in new Figure S3D. The data show somewhat reduced surface expression of the mutant DDR2 constructs, which aligns better with the comments about potential effects on cellular trafficking. The updated text is in the Results section, lines 396-398 and Discussion section, lines 687, 694 – 701.

The ADP-Glo assay used is an endpoint assay that measures the total phosphorylation happening after time “t”, not a kinetic continuous assay, which should work to assess specific activity for the enzyme. I am not sure if it can be used accurately to measure Michaelis Menten kinetic constants. The measured activity might not represent the initial rate if the reaction has progressed beyond the linear phase of the reaction progress curve, where the rate is constant. The methods section mentions that the reaction was stopped at 10, 20, and 40 minutes, the earliest of which (10 min) far exceeds the 2 minutes that it takes the mutant and FP protein to fully phosphorylate the A loop per the WB data, hence not measuring the initial rates for sure. Thus, Table 1, results, and discussion should only copare the specific activity of the constructs.

Response: The ADP-Glo assay is suitable to be used to determine Michaelis Menten kinetics because it measures ADP production over time (see also Zegzouti et al, Assay Drug Dev Technol. 2010). To clarify, we do not use this assay to measure autophosphorylation, as the reviewer seems to believe, but to measure phosphorylation of a model substrate peptide (Axltide peptide, also used in Iwai et al, Biochem J, 2013). We used the assay in the same way as we did in our previous study where we determined the Michale Menten kinetics for the DDR1 kinase (Sammon et al, PNAS, 2020). In both studies, ADP production was linear up to the last time point. This was checked in every assay. Furthermore, we have calculated ADP production at the highest activity (DDR2-KY740C incubated with 520 μM Axltide peptide) and found that only 21.6% of the peptide had been phosphorylated under these conditions.

I am not sure what Figure 4B adds that is different than Figure 3B (other than using 100-fold of the enzyme concentrations, which I cannot justify), since they both seem to compare the time course of in vitro autophosphorylation of DDR1 and DDR1. (Lines 447 and 448: Are these the same two constructs, and are these the same ones in line 456/457?). If all the same, I suggest merging the native gel panel with Figure 3 for a more concise figure.

Response: We have now given more rationale for using the two different concentrations of DDR constructs, lines 474 – 476. We considered merging the native gel with Figure 3 and adding the bottom panels of Figure 4 as a new Supplemental Figure. On careful reflection, we decided to keep the original Figures 3 and 4, as it would be difficult for the reader to follow the text (lines 476 – 487) that describes both the native gel and the Western blotting results if the Western blotting data were shown in a supplementary Figure, separate from the native gel.

In the discussion, the authors do a great job summarizing mutation effect in RTK, yet they need to acknowledge that because the constructs used lack some of the cytosolic components of the JM domain that were present in other studies, it’s harder to draw the conclusion that Src activation is not needed in a physiologically relevant environment.

Response: The Discussion has been expanded to include a comment about possible DDR2 activation by Src in cells, lines 649 – 653.

---

## [Decision Letter · Decision Letter 1]

2 Nov 2025

Warburg-Cinotti disease variant p.Tyr740Cys enhances catalytic activity of DDR2 kinase

PONE-D-25-34746R1

Dear Dr. Leitinger,

We’re pleased to inform you that your manuscript has been judged scientifically suitable for publication and will be formally accepted for publication once it meets all outstanding technical requirements.

Kind regards,

Jianhong Zhou

Staff Editor

PLOS ONE

Additional Editor Comments (optional):

Reviewers' comments:

Reviewer's Responses to Questions

**Comments to the Author**

Reviewer #1: (No Response)

Reviewer #2: All comments have been addressed

2. Is the manuscript technically sound, and do the data support the conclusions?

Reviewer #1: (No Response)

Reviewer #2: Yes

3. Has the statistical analysis been performed appropriately and rigorously?

Reviewer #1: (No Response)

Reviewer #2: Yes

4. Have the authors made all data underlying the findings in their manuscript fully available?

Reviewer #1: (No Response)

Reviewer #2: Yes

5. Is the manuscript presented in an intelligible fashion and written in standard English?

Reviewer #1: (No Response)

Reviewer #2: Yes

Reviewer #1: (No Response)

Reviewer #2: Thank you for addressing the comments. Please add the limitation to using ADP Global assay in determining the Michaelis Menten constant, the accuracy of the V0 is an approximation because the assay stops at a single point.

**Do you want your identity to be public for this peer review?** For information about this choice, including consent withdrawal, please see our Privacy Policy

Reviewer #1: No

Reviewer #2: No

---

## [Editor Report · Acceptance letter]

PONE-D-25-34746R1

PLOS ONE

Dear Dr. Leitinger,

I'm pleased to inform you that your manuscript has been deemed suitable for publication in PLOS ONE. Congratulations! Your manuscript is now being handed over to our production team.

Kind regards,

on behalf of

Dr. Jianhong Zhou

Staff Editor

PLOS ONE